

# The conformal spectrum of non-Abelian anyons

**Nima Doroud**[1]$^\star$**, David Tong**[2]$^\dagger$ **and Carl Turner**[2]$^\ddagger$

**1** International School for Advanced Studies (SISSA)
Via Bonomea 265, 34136 Trieste, Italy
**2** Department of Applied Mathematics and Theoretical Physics,
University of Cambridge, CB3 0WA, UK

$\star$ ndoroud@sissa.it        $\dagger$ d.tong@damtp.cam.ac.uk        $\ddagger$ c.p.turner@damtp.cam.ac.uk

## Abstract

We study the spectrum of multiple non-Abelian anyons in a harmonic trap. The system is described by Chern-Simons theory, coupled to either bosonic or fermionic non-relativistic matter, and has an $SO(2,1)$ conformal invariance. We describe a number of special properties of the spectrum, focussing on a class of protected states whose energies are dictated by their angular momentum. We show that the angular momentum of a bound state of non-Abelian anyons is determined by the quadratic Casimirs of their constituents.



## Contents

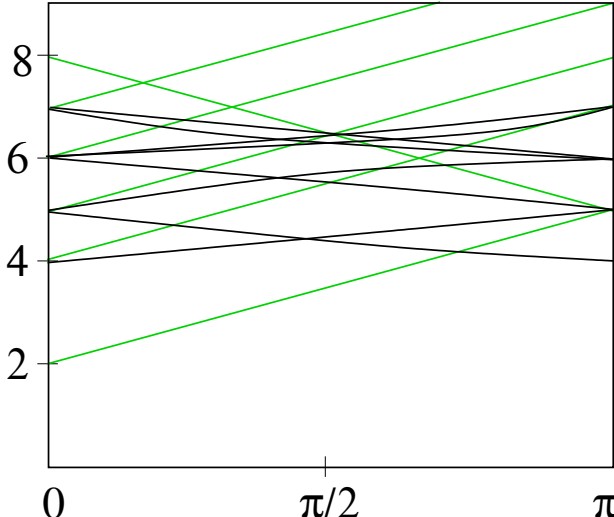

Figure 1: Spectrum of low-lying states for $n = 3$ anyons with repulsive interactions, computed numerically. Based on the results of [13].

# 1 Introduction and summary

The quantum mechanics of multiple, interacting anyons is a wonderfully rich problem. It is simple to state but contains a wealth of interesting physics. Despite several decades of interest, it remains unsolved. The purpose of this paper is to fail to solve the harder problem of interacting non-Abelian anyons.

In this extended introduction, we will first summarise the story of Abelian anyons. These are particles which, upon an anti-clockwise exchange, pick up a phase $e^{i\theta}$. We will write $\theta = \pi/k$ so that the anyons are bosons when $k = \infty$ and fermions when $k = 1$. In a field theoretic language, anyons are described by a $U(1)$ Chern-Simons theory at level $k$, coupled to a non-relativistic scalar field.

We will explore the spectrum of $n$ anyons placed in a harmonic trap. (See [1, 2] for early work on this subject, and [3–5] for reviews.) The trap has the potential

$$V = \frac{\omega^2}{2}(x^2 + y^2).$$

To fully specify the Hamiltonian, we also need to describe any interactions between the anyons. It turns out that the problem simplifies tremendously if the particles experience pairwise, contact interactions [6–10]. The strength of these interactions is determined by seeking a fixed point of an RG flow. However, the sign of the coupling is arbitrary. This leaves us with two options – attractive and repulsive interactions – exhibiting interesting and different physics.

As an aside, we should mention that when these contact interactions are turned on, the quantum mechanics has an $SO(2, 1)$ conformal invariance of the type first introduced in [11] and subsequently explored in [12]. This conformal invariance will play an important role throughout this paper but we will not focus on it for the rest of this introduction.

Perhaps the best way to illustrate the physics of anyons is simply to look at the spectrum. Low-lying states were computed numerically for $n = 3$ anyons with repulsive interactions by Sporre, Verbaarschot, and Zahed [13]. A sketch of their results is shown in Fig. 1. (A similar plot for $n = 4$ anyons can be found in [14].) The energy $E$ is plotted vertically[1] and the statistical parameter $\theta \in [0, \pi]$ is plotted horizontally. The spectrum on the far left coincides

---

[1]In terms of our conventions, the plot actually shows $E - \omega$, measured in units of $\omega$.

with that of free bosons; on the far right it coincides with free fermions. In between, things are more interesting.

This plot contains some things that are easy to understand and some things that are hard. Let's start with the hard. The most striking feature is that there is a level crossing of the ground state as $\theta$ is increased. Roughly speaking this occurs because the anyons have an intrinsic angular momentum that scales as $\theta$. As we increase $\theta$, we increase both the angular momentum and the energy of the state. For some value of $\theta$, both of these can be lowered if the particles start orbiting in the opposite direction to their intrinsic spin. This is where the ground state level crossing occurs. A similar level crossing is expected for all $n$, but little is known beyond these numerical results.

**Some simple states**

In contrast, some aspects of the spectrum are fairly easy to understand. In particular, there are a number of states whose energy varies linearly with $\theta$. Among these is the small-$\theta$ ground state, but not the large-$\theta$ ground state which takes over after the level crossing. For obvious reasons, these are sometimes referred to as "linear states" [15,16]. They persist in the spectrum of $n$ anyons and, in all cases, their wavefunctions and energies are known exactly. For example, in the $n$ anyon quantum mechanics with repulsive interactions, the ground state close to the bosonic end of the spectrum (i.e. for suitably large $k$) has energy

$$E = \left( n + \frac{n(n-1)}{2k} \right) \omega. \tag{1}$$

Here the first term is simply the ground state energy of $n$ particles in a two-dimensional harmonic trap (it is $2 \times \frac{1}{2} \hbar \omega$ for each particle, with $\hbar = 1$). The second term can be thought of as a correction due to the inherent angular momentum of the particles.

The fact that some states in the spectrum have such a simple expression for their energy strongly suggests that there is some underlying symmetry that protects them. Indeed there is: it is supersymmetry! This is particularly surprising given that the anyonic quantum mechanics does not have supersymmetry, but is nonetheless true. The reasoning starts with the observation that it possible to write down a supersymmetric theory of two species of anyons whose spins differ by 1/2 [17]. When restricted to states involving just one species of anyons, this reduces to our problem of interest. Such a statement would not be true in relativistic theories, in which particle-anti-particle pair creation prevents other fields from decoupling at the loop level. However, the lack of anti-particles means that it does hold in our non-relativistic theories. The supersymmetric theory of anyons has short, BPS multiplets whose energies are fixed in terms of their quantum numbers [18,19]. These BPS states coincide with the "linear states" in the anyon spectrum [20].

It's worth explaining in more detail how this arises. For $n$ anyons, the BPS states have energy given by

$$E = (n - J) \omega, \tag{2}$$

with $J$ the total angular momentum of $n$ anyons. One of the surprising properties of the angular momentum of anyons is that it does not add linearly. Instead, one finds that $J \sim n^2$ for large $n$, together with some sub-leading corrections which are more subtle and depend, even classically, on a choice of regularisation procedure [21–23]. (We will review this in some detail in later sections.) In the present context, a careful analysis shows that

$$J = -\frac{n(n-1)}{2k},$$

so that the BPS bound (2) indeed reproduces the energy spectrum (1).

**Non-Abelian anyons**

The purpose of this paper is to extend the discussion above to non-Abelian anyons. The simplest way to construct such particles is to couple fields to a non-Abelian Chern-Simons theory. For example, in Section 3, we will consider an $SU(N)_k$ Chern-Simons theory coupled to scalar fields. Each of these scalar fields transforms in some representation $R$ of $SU(N)$.

Suppose that we place $n$ non-Abelian anyons in a harmonic trap, each labelled by some representation $R_i$ with $i = 1, \ldots, n$. We once again tune the contact interactions so that the theory sits at an RG fixed point. Our goal is to understand the energy spectrum.

We will fall short of this goal. As with Abelian anyons, there are many questions that we are unable to answer analytically, such as those about possible level crossings in the ground state of the system. We will, however, show that there are states in the spectrum analogous to (1) whose energy can be determined exactly. We show that the energy of these states again takes the form $E = (n - J)\omega$ but this still leaves open the problem of determining the angular momentum $J$ of $n$ non-Abelian anyons. This is determined by some simple group theory.

Suppose, for example, that we place $n = 2$ anyons in a trap with representations $R_1$ and $R_2$. The possible representations of the resulting bound states are determined by the decomposition of the tensor product $R_1 \otimes R_2$. The angular momentum of the bound state in the irreducible representation $R \subset R_1 \otimes R_2$ turns out to be

$$J = -\frac{C_2(R) - C_2(R_2) - C_2(R_1)}{2k},\tag{3}$$

where $C_2(R)$ is the quadratic Casimir of the representation $R$. This, in turn, determines the energy of this state using (2). We will see that there is a straightforward generalisation of this result to $n$ anyons, each of which sits in a different representation.

The remainder of this paper is primarily devoted to telling the story above and providing a number of examples. The tools we will use are those of non-relativistic field theory, rather than non-relativistic quantum mechanics. In Section 2, we review the properties of field theories that enjoy a non-relativistic $SO(2,1)$ conformal symmetry. This conformal extension of the Galilean symmetry is known as the Schrödinger symmetry. The state-operator map in such theories allows us to translate the problem of the spectrum of anyons in a harmonic trap to the problem of computing the scaling dimension of certain operators.

In Section 3, we consider a bosonic Chern-Simons matter theory. Much of this section is devoted to proving the result (3) for the angular momentum of two anyons, as well as its generalisation to $n$ anyons. We use this to determine the energy of these states, and confirm our results with explicit one-loop computations. In Section 4 we repeat this story for fermionic Chern-Simons matter theories.

## 2 Non-relativistic conformal invariance

The purpose of this paper is to investigate the spectrum of non-Abelian anyons in a harmonic trap. The most natural setting to address this problem is Chern-Simons theory, where flux attachment and the associated Aharonov-Bohm effect give rise to the desired non-Abelian statistics.

The theories we will study have a non-relativistic conformal invariance. We will describe these theories in some detail in later sections. In this section, we start by reviewing some basic aspects of conformal invariance in non-relativistic field theories, following the seminal work of Nishida and Son [24].

For high-energy theorists, used to studying relativistic conformal field theories, some aspects of their non-relativistic counterparts can be a little counter-intuitive. In an attempt to

reorient these readers, we begin by stating the blindingly obvious. First, non-relativistic field theories, conformal or otherwise, describe the dynamics of massive particles. Second, these theories do not have anti-particles. This means that much of the subtlety of relativistic quantum field theory disappears. Indeed, if we choose to focus on a sector of a non-relativistic theory with a fixed particle number, then the theory reduces to quantum mechanics. Nonetheless, the field theoretic description is often more useful and, despite the very obvious differences described above, there are ultimately similarities between relativistic and non-relativistic conformal theories.

For simplicity, suppose that all particles have the same mass $m$. We introduce the particle density $\rho(\mathbf{x})$ and momentum density $\mathbf{j}(\mathbf{x})$, where we are working in the Schrödinger picture so that field theoretic operators do not depend on time. From these we can build the familiar conserved charges corresponding to particle number $\mathcal{N}$, total momentum $\mathbf{P}$ and angular momentum $J$:

$$\mathcal{N} = \int \mathrm{d}^2x \; \rho(\mathbf{x}) \,, \quad \mathbf{P} = \int \mathrm{d}^2x \; \mathbf{j}(\mathbf{x}) \,, \quad J = \int \mathrm{d}^2x \; \mathbf{x} \times \mathbf{j}(\mathbf{x}) \,.$$

As in any quantum system, time evolution is implemented by the Hamiltonian $H$. The continuity equation then reads

$$i[H, \rho] + \nabla \cdot \mathbf{j} = 0 \,.$$

In a conformal field theory, there are three further, less familiar, generators that we can also build from $\rho$ and $\mathbf{j}$. These are the generators of Galilean boosts $\mathbf{G}$, the dilatation operator $D$ and the special conformal generator $C$, defined as

$$\mathbf{G} = \int \mathrm{d}^2x \; \mathbf{x}\,\rho(\mathbf{x}) \,, \quad D = \int \mathrm{d}^2x \; \mathbf{x} \cdot \mathbf{j}(\mathbf{x}) \,, \quad C = \frac{m}{2} \int \mathrm{d}^2x \; \mathbf{x}^2 \rho(\mathbf{x}) \,. \tag{4}$$

To these we should add the Hamiltonian $H$. In a conformal field theory, these generators obey the algebra

$$i[D, \mathbf{P}] = -\mathbf{P} \,, \quad i[D, \mathbf{G}] = +\mathbf{G} \,, \quad i[D, H] = -2H \,, \quad i[D, C] = +2C \,,$$
$$i[C, \mathbf{P}] = -\mathbf{G} \,, \quad [H, \mathbf{G}] = -i\mathbf{P} \,, \quad [H, C] = -iD \,, \quad [P_i, G_j] = -im\mathcal{N}\delta_{ij} \,. \tag{5}$$

with all other commutators that don't involve $J$ vanishing. This is sometimes referred to as the Schrödinger algebra. The triplet of operators $H$, $D$ and $C$ form an $SO(2,1)$ subgroup. The commutators $[J, \mathbf{P}]$ and $[J, \mathbf{G}]$ are non-zero and tell us that $\mathbf{P}$ and $\mathbf{G}$ transform as vectors.

## 2.1 States and operators

In conformal theories, the spectrum of the Hamiltonian is necessarily continuous. Instead, as with their relativistic counterparts, the interesting questions lie in the spectrum of the dilatation operator $D$.

We consider local operators, evaluated at the origin: $\mathcal{O} = \mathcal{O}(\mathbf{x} = 0)$. These operators can be taken to have fixed particle number $n_\mathcal{O}$ and angular momentum $j_\mathcal{O}$, defined by

$$[J, \mathcal{O}] = j_\mathcal{O}\mathcal{O} \,, \quad [\mathcal{N}, \mathcal{O}] = n_\mathcal{O}\mathcal{O} \,.$$

Unitarity restricts $n_\mathcal{O} \geq 0$. This is the statement that there are no anti-particles in the theory.

More interesting are the transformations under dilatations. We say that the operators have *scaling dimension* $\Delta_\mathcal{O}$ if they obey

$$i[D, \mathcal{O}] = -\Delta_\mathcal{O}\mathcal{O} \,.$$

If we find one operator $\mathcal{O}$ with definite scaling dimension, then the algebra (5) allows us to construct an infinite tower of further operators with the same property. Both $H$ and $\mathbf{P}$ act as raising operators: $[H, \mathcal{O}]$ has scaling dimension $\Delta_{\mathcal{O}} + 2$ and $[\mathbf{P}, \mathcal{O}]$ has scaling dimension $\Delta_{\mathcal{O}} + 1$. In contrast, both $C$ and $\mathbf{G}$ act as lowering operators: $[C, \mathcal{O}]$ has scaling dimension $\Delta_{\mathcal{O}} - 2$ while $[\mathbf{G}, \mathcal{O}]$ has scaling dimension $\Delta_{\mathcal{O}} - 1$.

The spectrum of $D$ must be bounded below. Indeed, a simple unitarity argument [25] shows that

$$\Delta_{\mathcal{O}} \geq 1 . \tag{6}$$

This means that there must be operators sitting at the bottom of the tower which obey

$$[\mathbf{G}, \mathcal{O}] = [C, \mathcal{O}] = 0 .$$

Such operators are called *primary* [24,26]. The other operators in the tower are called *descendants*; they can be constructed by acting with $H$ and $\mathbf{P}$. The full tower built in this way is an irreducible representation of the Schrödinger algebra.

**The state-operator Map**

One of the most beautiful aspects of relativistic conformal field theories is the state operator map. This equates the spectrum of the dilatation operator on the plane to the spectrum of the Hamiltonian when the theory is placed on a sphere.

There is also such a map in non-relativistic conformal field theories which, if anything, is even more simple. First, the algebra: we define a modified Hamiltonian

$$L_0 = H + C . \tag{7}$$

For each local, primary operator $\mathcal{O}(0)$, we define the state $|\Psi_{\mathcal{O}}\rangle = e^{-H} \mathcal{O}(0)|0\rangle$. Then it is simple to check that

$$L_0 |\Psi_{\mathcal{O}}\rangle = \Delta_{\mathcal{O}} |\Psi_{\mathcal{O}}\rangle . \tag{8}$$

Further, $J|\Psi_{\mathcal{O}}\rangle = j_{\mathcal{O}}|\Psi_{\mathcal{O}}\rangle$ and $\mathcal{N}|\Psi_{\mathcal{O}}\rangle = n_{\mathcal{O}}|\Psi_{\mathcal{O}}\rangle$.

Now the physics: we view $L_0$ as a new Hamiltonian, with a very simple interpretation. This follows from the definition of $C$ in (4) which tells us that we have taken the original theory, defined by $H$, and placed it in a harmonic trap. (We have used conventions where the strength of the harmonic trap is $\omega = 1$.) The spectrum of particles in this harmonic trap is equal to the spectrum of the dilatation operator. This was first pointed out for field theories in [24], although the analogous statement in quantum mechanics can be traced back to the earliest work on conformal invariance [11].

In relativistic theories, we are very used to the state-operator map holding only for local operators. This limitation is usually thought to also hold in the non-relativistic framework considered here. However, in Section 3, we will see that we can also apply this map to certain Wilson line operators.

The tower of descendant operators maps into a tower of higher energy states in the trap. There are two ways to raise the energy. The first is to construct states which sit further out in the trap. This is achieved by introducing the raising and lowering operators

$$L_{\pm} = H - C \pm iD \quad \Rightarrow \quad \begin{cases} \left[L_0, L_{\pm}\right] = \pm 2L_{\pm} \\ \left[L_+, L_-\right] = -4L_0 \end{cases}$$

The second way is to take a given state and make it oscillate backwards and forwards. This is achieved by introducing the complexified momentum,

$$\mathscr{P} = \mathbf{P} + i\mathbf{G} \quad \Rightarrow \quad \begin{cases} \left[L_0, \mathscr{P}\right] = \mathscr{P} \\ \left[L_0, \mathscr{P}^\dagger\right] = -\mathscr{P}^\dagger \end{cases}$$

The primary states sit at the bottom of this tower and obey $L_- |\Psi_\mathcal{O}\rangle = \mathscr{P}^\dagger |\Psi_\mathcal{O}\rangle = 0$. Acting on these primary states with $L_+$ and $\mathscr{P}$ raises the energy, filling out the representation of the Schrödinger algebra.

## 3 The bosonic theory

In this section we study a class of $d = 2 + 1$ Chern-Simons-matter theories. For concreteness, we will take the gauge group to be $SU(N)_k$, where $k$ denotes the level, although everything we say generalises to arbitrary gauge groups. The Chern-Simons action takes the familiar form

$$S_{CS} = -\frac{k}{4\pi} \int \mathrm{d}^3 x \, \mathrm{Tr} \, \epsilon^{\mu\nu\rho} (A_\mu \partial_\nu A_\rho - \frac{2i}{3} A_\mu A_\nu A_\rho) \,. \tag{9}$$

The Chern-Simons theory is coupled to non-relativistic matter. In this section, this will take the form of $N_f$ scalar fields $\phi_a$, with $a = 1, \ldots, N_f$. Each of them transforms in some representation $R_a$ under $SU(N)$. We denote the corresponding generators as $t^\alpha[R_a]$ where $\alpha = 1, \ldots, N^2 - 1$ and we have suppressed the matrix indices. The generators in the fundamental representation are normalised such that $\mathrm{Tr} \, t^\alpha t^\beta = \delta^{\alpha\beta}$. Each of these scalar fields is endowed with a non-relativistic kinetic term. For simplicity, we give each particle the same mass $m$. The action is given by

$$S = \int \mathrm{d}t \, \mathrm{d}^2 x \, \left\{ i\phi_a^\dagger \mathscr{D}_0 \phi_a - \frac{1}{2m} \vec{\mathscr{D}} \phi_a^\dagger \vec{\mathscr{D}} \phi_a - \lambda (\phi_a^\dagger t^\alpha[R_a] \phi_a)(\phi_b^\dagger t^\alpha[R_b] \phi_b) \right\} \,. \tag{10}$$

The quartic term gives rise to a delta-function interaction between particles. The coupling $\lambda$ is marginal and is known to run logarithmically. There are two fixed points given by [7,8]

$$\lambda = \pm \frac{\pi}{mk} \,.$$

The $\lambda > 0$ fixed point is stable; the $\lambda < 0$ fixed point is unstable. In what follows, we choose to set

$$\lambda = +\frac{\pi}{mk} \,, \tag{11}$$

but we do not fix the sign of the Chern-Simons coupling $k$, so this choice includes both stable and unstable fixed points.

   This fixed point also exists in the $U(1)$ theory, where $\lambda > 0$ corresponds to repulsive interactions between particles and $\lambda < 0$ corresponds to attractive interactions. In the non-Abelian theory, this classification is not so simple because, for a fixed sign of $\lambda$, interactions in channels for different irreducible representations $R \subset R_1 \otimes R_2$ can be either attractive or repulsive. (Such behaviour also holds in classical Yang-Mills theory. For example, a quark and anti-quark attract in the singlet channel, but repel in the adjoint channel.)

   It has long been known that the fixed points (11) exhibit an enhanced non-relativistic conformal invariance of the sort described in Section 2 [2]. The generators of the conformal algebra are constructed from the particle density and momentum current

$$\rho = \phi_a^\dagger \phi_a \quad \text{and} \quad \mathbf{j} = -\frac{i}{2} \left( \phi_a^\dagger \vec{\mathscr{D}} \phi_a - (\vec{\mathscr{D}} \phi_a^\dagger) \phi_a \right),$$

together with the Hamiltonian

$$H = \int \mathrm{d}^2 x \ \frac{2}{m} \mathcal{D}_{\bar{z}} \phi_a^\dagger \mathcal{D}_z \phi_a \,, \tag{12}$$

where we have introduced complex coordinates on the plane $z = x^1 + i x^2$ .

Our real interest lies in the spectrum of non-Abelian anyons when placed in a harmonic trap. In the present context, this means that we want the spectrum of $L_0 = H + C$. As we explained in Section 2, this is equivalent to determining the spectrum of the dilatation operator $D$. It turns out that this latter formulation of the problem is somewhat simpler to work with.

### 3.1 Gauge invariant operators

The first thing to do is to identify the operators of interest. As always, we must talk about gauge invariant operators. We will construct such operators simply by attaching Wilson lines stretching out to infinity. Thus we define

$$\Phi_a(\mathbf{x}) = \mathscr{P} \exp\left( i \int_\infty^{\mathbf{x}} A^\alpha \, t^\alpha[R_a] \right) \phi_a(\mathbf{x}) \,. \tag{13}$$

This requires some explanation. $\Phi(\mathbf{x})$ is not a local operator; it depends on the value of the gauge field along a line stretching to infinity. Meanwhile, the state-operator map described in the previous section is usually taken to hold only for local operators. However, closer inspection of the argument leading to (8) shows that we require only that the operator $\mathcal{O}(\mathbf{x})$ has a well defined scaling dimension. It is simple to check that the Wilson line does not affect this property of $\Phi$.

Under the state operator map, the state $|\Phi_a^\dagger\rangle$ describes a single anyon, transforming in representation $R_a$, sitting in a harmonic trap. The particle retains the attached Wilson line and is entirely analogous to the correct description of a physical electron in QED. Importantly, the $SU(N)_k$ Chern-Simons theory does not confine and so this particle has finite energy. We will compute this energy explicitly below.

It's worth pausing to comment that the situation differs from that in relativistic conformal theories, where the state-operator map is restricted to local operators. Indeed, in the relativistic context the states are considered on a spatial sphere where there is no option to attach a Wilson line that stretches to infinity. Instead, in Chern-Simons theories Gauss' law requires that charged states are accompanied by monopole operators, which places further constraints on the possible electric excitations. At least for this aspect of the physics, thinking about Chern-Simons-matter theories with relativistic conformal invariance does not appear to be a good guide to the non-relativistic theories.

Now we can discuss the kinds of operators that we are interested in. In the $n$-particle sector, we will look at operators of the form

$$\mathcal{O} \sim \prod_{i=1}^n (\partial^{l_i} \bar{\partial}^{m_i} \Phi_{a_i}^\dagger) \,, \tag{14}$$

where we have introduced (anti-)holomorphic spatial derivatives $\partial = \frac{1}{2}(\partial_1 - i \partial_2)$ and $\bar{\partial} = \frac{1}{2}(\partial_1 + i \partial_2)$. The primary operators are those which cannot be written as a total derivative.

Before we proceed, a comment is in order. The operators written above are not the most general and, indeed, do not necessarily have fixed scaling dimension. This is because there's nothing to stop these from mixing with operators of the form $(\Phi^\dagger)^{n+p} \Phi^p$, possibly with derivatives attached too. However, because non-relativistic theories contain no anti-particles, these additional operators annihilate the vacuum $|0\rangle$ and so result in the same state $|\mathcal{O}\rangle$ under the

state-operator map. Since our real interest lies in the theory with the harmonic trap, for many purposes it will suffice to use (14) as a way to characterise the operators.

It is not a totally trivial task to list the primary operators from (14). The only one that is simple to write down has no derivatives

$$\mathcal{O}_{a_1 \ldots a_n} = \Phi^{\dagger}_{a_1} \ldots \Phi^{\dagger}_{a_n} \; . \tag{15}$$

(Here $a_i$ are flavour indices. We have suppressed colour indices.) The $n$-particle ground state is expected to take such a form for suitably large $k$; we will compute its energy shortly.

To highlight how other primary operators arise, it will be useful to look at a simple example. We take $U(1)_k$ with a single field $\phi$ of charge $+1$. (This was the case discussed in [20].) To make contact with the introduction and, in particular, the numerical spectrum of [13], let us look at the case $n = 3$. As we mentioned above, the large $k$ ground state is simply the state corresponding to $(\Phi^{\dagger})^3$, as we will see shortly through explicit computation. What about higher states? Any state with a single derivative can be written as a total derivative and so is a descendant. This explains the gap between the ground state and the first excited state seen in Figure 1. The next primary operator will contain two derivatives. There are six such operators: $\partial\Phi^{\dagger}\partial\Phi^{\dagger}\Phi^{\dagger}$, $\partial\Phi^{\dagger}\bar{\partial}\Phi^{\dagger}\Phi^{\dagger}$, $\bar{\partial}\Phi^{\dagger}\bar{\partial}\Phi^{\dagger}\Phi^{\dagger}$, $\partial^2\Phi^{\dagger}\Phi^{\dagger 2}$, $\partial\bar{\partial}\Phi^{\dagger}\Phi^{\dagger 2}$ and $\bar{\partial}^2\Phi^{\dagger}\Phi^{\dagger 2}$. However, four linear combinations of these can be written as total derivatives of the form $\partial(\partial\Phi^{\dagger}\Phi^{\dagger 2})$, where either derivative could also be $\bar{\partial}$. The upshot is that there are two primary states with two derivatives. This agrees with the spectrum shown in Figure 1.

We can play a similar game with operators that contain three derivatives. It is simple to check that one can write down 13 such operators, 10 of which turn out to be descendants. The upshot is that there are 3 primary operators that contain 3 derivatives. (The obvious pattern does not persist!) From Figure 1, we learn that one of these will become the ground state at small $k$.

## 3.2 The spectrum

Next comes the question that we initially set out to answer: what is the spectrum of the states (14)? As we stressed in the introduction, this is a difficult and unsolved question, even for Abelian anyons. Here we offer two approaches.

In Section 3.4 we explain how one can compute the spectrum of these operators for Chern-Simons theory with scalars perturbatively in $1/k$. We present the results only at one-loop.

However, before we do this, there is a special class of operators for which the result simplifies tremendously. These are the "linear states" referred to in the introduction. They correspond to so-called *chiral operators* which have no anti-holomorphic derivatives,

$$\mathcal{O} \sim \prod_{i=1}^{n} (\partial^{m_i} \Phi^{\dagger}_{a_i}) \; . \tag{16}$$

The simplest such operator is $\mathcal{O}_{a_1 \ldots a_n}$ in (15). The scaling dimension of any such operator turns out to be fixed by its angular momentum $J$:

$$\Delta = n - J \; . \tag{17}$$

We will not explain the argument behind this here since it involves a detour through the supersymmetry algebra. Instead we refer the reader to our earlier paper [20] (which follows in the footsteps of [18, 19]) where this result is derived.[2]

---

[2] Our conventions differ slightly from those of [20], where we defined the angular momentum with an appropriate twist of the R-symmetry, so that $J = -n^2/2k$ for $n$ anyons. The convention chosen in this paper is more natural in the context of non-Abelian gauge theories.

Note that each derivative $\partial$ decreases the angular momentum by one. Correspondingly, the dimension of a chiral operator (16) is given by

$$\Delta_{\mathcal{O}} = n + \sum_{i=1}^{n} m_i - J_0 \, ,$$

where $J_0$ is the angular momentum of $\mathcal{O}_{a_1 \ldots a_n}$.

### 3.3 Angular momentum

From the discussion above, we learn that the dimension of $\mathcal{O}_{a_1 \ldots a_n}$ and other chiral operators (16) is entirely determined by the angular momentum $J$. But what is this angular momentum?

The tensor product of representations $\otimes_{i=1}^{n} R_{a_i}$ is decomposed into irreps. (Note that, despite the presence of the Chern-Simons term, it is the tensor product and not the fusion rules which are relevant here.) When the operator $\mathcal{O}$ sits in the representation $R$, its angular momentum is given by

$$J_0 = -\frac{C_2(R) - \sum_i C_2(R_{a_i})}{2k} \, , \tag{18}$$

where $C_2$ is the quadratic Casimir, defined by

$$\sum_{\alpha} t^{\alpha}[R] t^{\alpha}[R] = C_2(R)\mathbf{1} \, .$$

Note that because $\mathcal{O}_{a_1 \ldots a_n}$ is built out of commuting scalar fields $\Phi$, in the absence of any derivatives it must transform in the fully symmetrised representation $R_{\mathrm{sym}} = \mathrm{Sym}\left[\otimes_{i=1}^{n} R_{a_i}\right]$. However, the more general operators (16) can transform in other representations.

The purpose of this section is to prove the result (18). Before we do this, we will first look at some examples.

**Examples**

We start with an Abelian gauge theory $U(1)_k$, where representations are labelled by charge $q \in \mathbb{Z}$. The quadratic Casimir in this case is simply $C_2(q) = q^2$. The result (18) says that the angular momentum of $n$ anyons, each of charge 1, is given by

$$J = -\frac{n(n-1)}{2k} \, . \tag{19}$$

This is indeed the angular momentum of $n$ anyons. Moreover, when substituted into (17), it gives us the correct answer for the dimension of the $n$ anyon operator; this is the result quoted in (1).

Next, consider $SU(2)_k$. Representations of $SU(2)$ are labelled by a spin $s \in \frac{1}{2}\mathbb{Z}$ and the final bound state has spin $S = \sum_i s_{a_i}$. In this case, the angular momentum is given by

$$J = -\frac{S(S+1) - \sum_i s_{a_i}(s_{a_i} + 1)}{k} \, .$$

This has a simple generalisation to $n$ anyons, each of which sits in the fundamental representation of $SU(N)$. We have $C_2(\mathbf{N}) = (N^2 - 1)/N$. The bound state transforms in the $n^{\mathrm{th}}$ symmetric representation of $SU(N)$, with $C_2(\mathrm{Sym}^n(\mathbf{N})) = n(N-1)(N+n)/N$. We have

$$J = -\frac{n(n-1)}{2k} \times \frac{N-1}{N} \, . \tag{20}$$

Finally, consider a general representation of $SU(N)$ whose Young tableau has rows of length[3] $\lambda_1 \geq \lambda_2 \geq \cdots \geq \lambda_N$ has quadratic Casimir given by the formula $C_2(\lambda) = \langle \lambda, \lambda + 2\rho \rangle$, where $\lambda$ is the highest weight and $\rho$ is the Weyl vector. In particular, we have

$$C_2(\lambda) = \begin{cases} \sum_{i=1}^{N} \left[ \lambda_i^2 + (N + 1 - 2i)\lambda_i \right] & \text{for } U(N) \\ \sum_{i=1}^{N} \left[ \lambda_i^2 + (N + 1 - 2i)\lambda_i \right] - \frac{1}{N} \left( \sum_{i=1}^{N} \lambda_i \right)^2 & \text{for } SU(N) \end{cases}$$

This translates into the following result for $n$ fundamental anyons brought together into the representation $\lambda$:

$$J = -\frac{\sum_{i=1}^{N} \left[ \lambda_i^2 - (2i - 1)\lambda_i \right]}{2k} \quad \text{for } U(N) ,$$

and

$$J = -\frac{\sum_{i=1}^{N} \left[ \lambda_i^2 - (2i - 1)\lambda_i \right] - n(n - 1)/N}{2k} \quad \text{for } SU(N) .$$

**Deriving the angular momentum**

We now return to prove the result (18) for the angular momentum. We insert $n$ anyons in various representations $R_{a_i}$ of the group $G$ at level $k$, such that they collectively transform in the irrep $R \subset \otimes_i R_{a_i}$. There are two issues which we need to explain. The first is that angular momentum of this state, inserted at the origin, is related to the quadratic Casimir $C_2(R)$. The second is that there are some ambiguities to do with regulators, but that the correct choice of angular momentum for our purposes is the one given above:

$$J = -\frac{C_2(R) - \sum_i C_2(R_{a_i})}{2k} .$$

It will be helpful to first develop some intuition for how this quadratic behaviour arises. It can be understood by considering the phase of the wavefunction for our $n$ anyons under rotations. To see this, place each anyon at a different distance from the origin. Now rotate the configuration by $2\pi$. In doing this, each anyon encircles all the others which are closer to the origin than itself, accumulating an Aharonov-Bohm phase per pair of particles. We additionally pick up a phase due to the inherent spin of each individual anyon. As we scale the configuration towards the origin, these are the only phases contributing to the behaviour of the wavefunction.

This decomposition into two phases is very similar to the usual decomposition of angular momentum into orbital and spin parts. We will find that the $J$ arising in the conformal (and superconformal) algebra is the one without intrinsic spins.

We now gather the ingredients needed for the computation. The angular momentum used in our algebra is given by

$$J = \int d^2z \sum_a \phi_a^\dagger (z \mathcal{D}_z - \bar{z} \mathcal{D}_{\bar{z}}) \phi_a .$$

To begin with, since we are going to place all particles at the origin, we can ignore the normal orbital angular momentum terms $\phi^\dagger (z \partial_z - \bar{z} \partial_{\bar{z}}) \phi$. Further, we will compute the angular momentum of states satisfying Gauss' law which, for our bosonic theory, reads

$$B^\alpha = \frac{2\pi}{k} \sum_a \phi_a^\dagger t^\alpha \phi_a , \tag{21}$$

---

[3]Note that by including the possibility of $\lambda_N \neq 0$, this formula works even for baryons. $\lambda_N$ has the interpretation of the number of baryons in the state.

where $B^\alpha = F_{12}^\alpha$ is the non-Abelian magnetic field. We then have

$$J = \frac{ik}{2\pi} \int d^2z \, (\bar{z}A_{\bar{z}}^\alpha - zA_z^\alpha)B^\alpha \, ,$$

acting on a state satisfying Gauss' law.

The reason for doing this is that this expression is now only sensitive to the Wilson line in (13). Let us give this a name: pick some representation $t^\alpha$, and let

$$W(\mathbf{x}) = \left[ \mathscr{P} \exp\left( i \int_\infty^{\mathbf{x}} A^\alpha \, t^\alpha \right) \right]^\dagger \, .$$

If Gauss' law (21) holds for the object $\Phi = W^\dagger \phi$, then it is straightforward to show that[4]

$$\left[ \frac{k}{2\pi} B^\alpha(\mathbf{x}), W(\mathbf{x}') \right] = W(\mathbf{x}') \, t^\alpha \, \delta^{(2)}(\mathbf{x} - \mathbf{x}') \tag{22}$$

This is enough to start computing the action of $J$ on a state containing Wilson lines. We take the Wilson lines to be $W_i = W(z_i, \bar{z}_i)\big|_{t=t_i}$, where we allow each $W_i$ to sit in a different representation $R_{a_i}$ whose generators are $t_{a_i}^\alpha$. Explicitly,

$$J \, W_1 \otimes \cdots \otimes W_n \, |0\rangle = \frac{ik}{2\pi} \int d^2z \, (\bar{z}A_{\bar{z}}^\alpha - zA_z^\alpha)B^\alpha \, W_1 \otimes \cdots \otimes W_n \, |0\rangle$$

$$= i \int d^2z \, (\bar{z}A_{\bar{z}}^\alpha - zA_z^\alpha)\Big[ W_1 \otimes \cdots \otimes W_n \Big] \sum_{i=1}^n t_{a_i}^\alpha \, \delta^{(2)}(z - z_i) \, |0\rangle \, ,$$

where $t_{a_i}$ is understood to act only on the $i^{\text{th}}$ factor of the product to the left.

Now we set $\mathbf{x}' = 0$ in (22) and use the complex coordinate $z = x_1 + ix_2 = re^{i\theta}$. If we integrate over a disc of radius $r$, the integral reduces to a boundary term, and

$$\frac{1}{2\pi} \int d\theta \left[ \bar{z}A_{\bar{z}}^\alpha(z,\bar{z}) - zA_z^\alpha(z,\bar{z}), W(0) \right] = \frac{i}{k} W(0) \, t^\alpha \, .$$

That is, evaluating this quantity around a circle gives this particular non-zero contribution if the Wilson line ends inside that circle; by contrast, it is zero if the end is outside that circle. This shows why we need to be careful with regularisation.

To regularise, let us proceed as above by smearing each $W_i$ around progressively smaller circles, of radius $|z_1| > |z_2| > \cdots > |z_n|$, and then taking the smallest one to zero first. In this manner, we find that we get only *one* contribution to the result per distinct pair $(i, j)$. However, it is not yet clear what happens when both terms in $J$ hit the same Wilson line.

To address this last case, we need one final argument. The simplest line of reasoning is that any translationally invariant regularisation of terms like $[A_z(z_i, \bar{z}_i), W(z_i, \bar{z}_i)]$ must vanish if we multiply it by $z_i$ and then take $z_i \to 0$.[5]

---

[4]One can prove this using the commutation relation $[A_1^\alpha(\mathbf{x}), A_2^\beta(\mathbf{x}')] = -\frac{2\pi i}{k} \delta^{\alpha\beta} \delta^{(2)}(z - z')$ . arising from the term $-\frac{k}{4\pi} \text{Tr} \, \epsilon^{\mu\nu\rho} A_\mu \partial_\nu A_\rho$ in the Chern-Simons action $S_{\text{CS}}$. Notice that $ikB^\alpha/2\pi$ generates spatial gauge transformations: for any function $h^\alpha(\mathbf{x})$

$$\left[ \int d^2x \, \frac{ik}{2\pi} B^\alpha h^\alpha, A_m^\beta(\mathbf{x}') \right] = \mathscr{D}_m h^\beta(\mathbf{x}') \, .$$

But the Wilson line is charged only at its endpoints, and for compactly supported $h$ it transforms at the $\mathbf{x}$ end so that $\left[ \int d^2x \, \frac{ik}{2\pi} B^\alpha h^\alpha, W(\mathbf{x}') \right] = +iW(\mathbf{x}') \, t^\alpha h^\alpha(\mathbf{x}')$. Setting $h$ to be a delta function, we obtain the above result.

[5]For a more careful approach, one could work in holomorphic gauge $A_{\bar{z}} = 0$, and then solve explicitly for $A_z$ as an integral of $B$; similarly in the Abelian case Coulomb gauge would work. In these formalisms, one finds that $J$ can be expressed as a double integral of $B^\alpha(\mathbf{x})B^\alpha(\mathbf{x}')$ against a term like $\mathbf{x} \times \nabla \log(\mathbf{x} - \mathbf{x}')$. Then the self-interaction terms we are concerned with vanish for regularisations preserving reflections at arbitrary locations.

Now we have the result

$$J \, W_1 \otimes \cdots \otimes W_n \, |0\rangle = -\frac{1}{k} W_1 \otimes \cdots \otimes W_n \left[ \sum_{i<j} t^\alpha_{a_i} \otimes t^\alpha_{a_j} \right] |0\rangle \, ,$$

where as above, $t_{a_i}$ is understood to act only on the $i^{\text{th}}$ factor of the product to the left.

All that remains is to relate this to the quadratic Casimir. But notice that the product representation $R$ in which the particles sit has the generators $T^\alpha = \sum t^\alpha_{a_i}$, and hence the quadratic Casimir is given by

$$C_2(R) = T^\alpha T^\alpha = \sum_{i,j} t^\alpha_{a_i} \otimes t^\alpha_{a_j} = 2 \sum_{i<j} t^\alpha_{a_i} \otimes t^\alpha_{a_j} + \sum_i C_2(R_{a_i}) \, .$$

This completes our proof. We have

$$J = -\frac{C_2(R) - \sum_i C_2(R_{a_i})}{2k}$$

as claimed.

### 3.4 Perturbation theory

The field theoretic approach to non-relativistic anyons comes equipped with the powerful methods developed for relativistic field theories. In particular, we can use Feynman diagrams to compute quantum corrections order by order in perturbation theory.

The diagrammatic method applies in much the same way as for relativistic theories with one crucial difference: there are no anti-particles, not even in loops! The absence of propagating anti-particles drastically reduces the number of diagrams, rendering the loop computations tractable.

It also follows that, in theories with multiple flavours, one type of species does not affect the dynamics of the other types unless it is present as an external particle. This was particularly useful in [20] where we embedded the bosonic theory in a supersymmetric theory [17] which includes both bosons and fermions. The supersymmetry algebra shows that the scaling dimensions of certain chiral operators involving only bosons are one-loop exact, a property which then survives even when the fermions are thrown away! Indeed, this is the trick that allowed us to determine the dimensions (17) in terms of the angular momentum.

Our goal in this section is to determine the one-loop corrections to the dimensions of operators of the form (14). We will use this to confirm our previous, algebraic results (17) and (18).

**One-loop corrections**

The Feynman rules for (10) are a simple generalization of those presented in [20]. To avoid clutter we only discuss the case with only one species of scalars living in some representation $R$ of the gauge group. The generalisation to multiple species living in different representations is straightforward. We will use Greek letters $\rho, \sigma = 1, \dots, \dim(R)$ to denote the colour indices.

Since the interactions are at most quartic, all corrections arise from pairwise diagrams. Thus it suffices to compute the one-loop correction to the two-anyon operator

$$\partial^{n_1} \bar{\partial}^{m_1} \phi^\dagger_{\rho_1} \partial^{n_2} \bar{\partial}^{m_2} \phi^\dagger_{\rho_2} \, . \tag{23}$$

Note that we have not included the Wilson lines in this operator as they do not play any role in what follows. The one-loop corrections to (23) are encoded in the correlation function

$$\langle \phi_{\sigma_1}(p_1) \phi_{\sigma_2}(p_2) \, \partial^{n_1} \bar{\partial}^{m_1} \phi^\dagger_{\rho_1} \partial^{n_2} \bar{\partial}^{m_2} \phi^\dagger_{\rho_2} \rangle \, . \tag{24}$$

At tree level we schematically denote this correlation function by the diagram:

$$
\begin{aligned}
&= \delta^{\rho_1}_{\sigma_1}\delta^{\rho_2}_{\sigma_2}(-ip_{1z})^{n_1}(-ip_{1\bar{z}})^{m_1}(-ip_{2z})^{n_2}(-ip_{2\bar{z}})^{m_2} \\
&\quad + \delta^{\rho_1}_{\sigma_2}\delta^{\rho_2}_{\sigma_1}(-ip_{1z})^{n_2}(-ip_{1\bar{z}})^{m_2}(-ip_{2z})^{n_1}(-ip_{2\bar{z}})^{m_1} .
\end{aligned}
\tag{25}
$$

Henceforth we shall suppress the diagram labels. At one-loop order the four-point function (24) is corrected by the diagrams

$$
= -\frac{1}{2k}\log\frac{\Lambda}{\mu}\left(t^\alpha_{\rho_1\sigma_1}t^\alpha_{\rho_2\sigma_2}+t^\alpha_{\rho_1\sigma_2}t^\alpha_{\rho_2\sigma_1}\right)\left(-\frac{i}{2}P^+_z\right)^{n_1+n_2}\left(-\frac{i}{2}P^+_{\bar{z}}\right)^{m_1+m_2}+\mathcal{O}(\Lambda^2),
$$

with $P^\pm = p_1 \pm p_2$, and

$$
= \frac{1}{2k}\log\frac{\Lambda}{\mu}\left(t^\alpha_{\rho_1\sigma_1}t^\alpha_{\rho_2\sigma_2}K(p_1,p_2)+t^\alpha_{\rho_1\sigma_2}t^\alpha_{\rho_2\sigma_1}K(p_2,p_1)\right)+\mathcal{O}(\Lambda^2),
\tag{26}
$$

where $K(p_1,p_2)$ is given by the following integral over the Feynman parameter $x$:

$$
\begin{aligned}
K(p_1,p_2) = (-i)^l\int_0^1 dx\Bigg\{&\prod_{i=1}^{2}\left(\frac{P^+_z}{2}-\frac{(-1)^i x P^-_z}{2}\right)^{n_i}\frac{\partial}{\partial x}\left[\prod_{i=1}^{2}\left(\frac{P^+_{\bar{z}}}{2}-\frac{(-1)^i x P^-_{\bar{z}}}{2}\right)^{m_i}\right] \\
&-\frac{\partial}{\partial x}\left[\prod_{i=1}^{2}\left(\frac{P^+_z}{2}-\frac{(-1)^i x P^-_z}{2}\right)^{n_i}\right]\prod_{i=1}^{2}\left(\frac{P^+_{\bar{z}}}{2}-\frac{(-1)^i x P^-_{\bar{z}}}{2}\right)^{m_i}\Bigg\}
\end{aligned}
\tag{27}
$$

with $l = n_1 + n_2 + m_1 + m_2$.

Note that we only need the logarithmic correction to extract the contribution to the anomalous dimension. Moreover, the results above are all one needs to evaluate the anomalous dimension of operators of the form (14) at one-loop.

We remark that the operators of the form (14) may not have a well-defined dimension at one-loop.[6] Nonetheless for a special class of such operators it is an easy task to find operators with well-defined scaling dimensions. These are the chiral operators discussed earlier (16) which only include holomorphic derivatives. Below we consider a few examples of such operators.

**Examples**

As a warm-up, consider the following operator in an Abelian theory with a single species of unit charge scalars:

$$
\mathcal{O}_n = \phi^{\dagger n} .
\tag{28}
$$

This is the $n$-anyon operator with no derivatives. The kinematical factor in (26) vanishes and we only have the bubble diagrams (26) to sum over. As was discussed earlier, there is one such diagram for each pair, and each yields the same contribution

$$
= -\frac{1}{k}\log\frac{\Lambda}{\mu}+\mathcal{O}(1) .
$$

---

[6]As is evident from the results above, a two-anyon operator with fixed $n_i$ and $m_i$ mixes at one-loop with operators with the same number of derivatives. Furthermore the one-loop diagrams above have polynomial divergences which need to be removed by counter-terms with fewer derivatives.

This results in the anomalous dimension

$$\Delta - n = \frac{n(n-1)}{2k} = -J \, ,\tag{29}$$

in agreement with (19).

Now consider the operator $\mathcal{O}_{\rho_1 \ldots \rho_n} = \phi^\dagger_{\rho_1} \ldots \phi^\dagger_{\rho_n}$ in an $SU(N)$ Chern-Simons theory coupled to a single species of scalars in the fundamental representation of the gauge group. As in the previous examples the absence of derivatives implies that we only need to evaluate the bubble diagram:

$$\text{[diagram]} = -\frac{N-1}{2Nk} \log \frac{\Lambda}{\mu} \left( \delta^{\rho_1}_{\sigma_1} \delta^{\rho_2}_{\sigma_2} + \delta^{\rho_1}_{\sigma_2} \delta^{\rho_2}_{\sigma_1} \right) + \mathcal{O}(1) \, ,$$

where we have used the following identity satisfied by the generators of $SU(N)$ in the fundamental representation

$$t^\alpha_{\rho_1 \sigma_1} t^\alpha_{\rho_2 \sigma_2} = \delta^{\rho_1}_{\sigma_2} \delta^{\rho_2}_{\sigma_1} - \frac{1}{N} \delta^{\rho_1}_{\sigma_1} \delta^{\rho_2}_{\sigma_2} \, .\tag{30}$$

Taking the contribution from each pair into account we obtain the anomalous dimension

$$\Delta - n = \frac{n(n-1)(N-1)}{2Nk} \, ,\tag{31}$$

reproducing (20).

As our last example, consider the $SU(N)$ theory with one species of scalars in the fundamental representation. This time we look at the operator $\mathcal{O} = \phi^\dagger_{[\rho_1} \partial \phi^\dagger_{\rho_2} \ldots \partial^{n-1} \phi^\dagger_{\rho_n]}$. Contrary to the previous examples, the corrections to this operator only arise from the gluon exchange diagrams:

$$\text{[diagram]} = \frac{N+1}{2Nk} \log \frac{\Lambda}{\mu} \delta^{\rho_i}_{[\sigma_i} \delta^{\rho_j}_{\sigma_j]} \left[ (-ip_{iz})^{n_i} (-ip_{jz})^{n_j} - (-ip_{iz})^{n_j} (-ip_{jz})^{n_i} \right] + \mathcal{O}(\Lambda^2) \, .$$

The contribution is the same for every pair yielding the one-loop corrected dimension

$$\Delta_n = \frac{n(n+1)}{2} - \frac{n(n-1)}{2} \frac{N+1}{kN} \, ,$$

and in particular the dimension of the baryon operator is given by

$$\Delta_N = \frac{N(N+1)}{2} - \frac{N^2-1}{2k} \, .\tag{32}$$

Evaluating the angular momentum, remembering to include a contribution of $-1$ per $\partial$ derivative, yields the same result.

To illustrate this, in Fig. 2 we have plotted the low-lying states for two anyons in $SU(2)_k$ Chern-Simons theory coupled to scalars in the fundamental representation. The energy $E$ (in units of $\omega$) is plotted vertically and the statistical parameter[7] $\theta \in [0, \pi]$ is plotted horizontally. Here the black lines depict the linear states while the blue curves approximate the non-linear states up to $O(\theta^2)$.

It is straightforward to count the low-lying states for two $SU(2)$ anyons. In this case there are only two possible representations $R$ for the composite operator. These are the trivial representation (anti-symmetric) and the adjoint representation (symmetric) of $SU(2)$. In the

---

[7]The statistical parameter for the $SU(2)_k$ anyons is given by $\theta = 2\pi/k$.

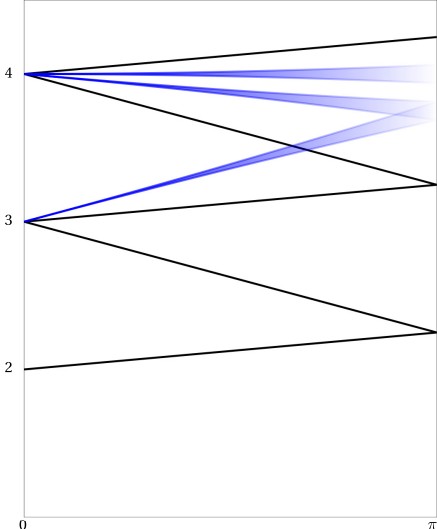

Figure 2: SU(2) anyons.

absence of any derivatives we have a single operator in the adjoint representation corresponding to the ground state. Allowing for a single derivative we find two descendants of the ground state as well as two primary operators only one of which corresponds to a linear state. Lastly there are twelve operators with two derivatives. Half of these are descendants and the other half are primary operators but only two of them give rise to linear states. Note that, with only two anyons in the trap, there is no level crossing for the ground state.

To find level crossing, we need to turn to three or more $SU(2)$ anyons. However, as we increase the number of anyons, the number of derivatives, or the rank of the gauge group, the counting of operators becomes more involved. In general, we do not have a systematic way to determine the number of chiral, primary and descendant operators. It would be interesting to better understand this counting of states.

## 3.5   Operators at the unitarity bound

For Abelian anyons, there is a rather striking difference between repulsive interactions, with $k > 0$, and attractive interactions, with $k < 0$. In both cases, the dimension of the BPS state $\mathcal{O} \sim \Phi^{\dagger n}$ is given by

$$\Delta = \left( n + \frac{n(n-1)}{2k} \right) .$$

For $k > 0$, this is the spectrum of anyons discussed in the introduction. For $k < 0$, there is a new twist to the story because, for $n > |2k|$, this operator appears to violate the unitarity bound $\Delta \geq 1$.

This issue was resolved in [20]. (Similar issues were also addressed in [27].) The presence of attractive delta-function interactions between anyons mean that the wavefunctions diverge as particles come close. For a suitably small number of particles, the divergence in the state $\mathcal{O} \sim \Phi^{\dagger n}$ is normalisable and the wavefunction describes an honest state in the Hilbert space. However, when the number of particles hits $n = 2k$, the divergence becomes logarithmically non-normalisable and this state is no longer part of the Hilbert space. The true ground state now requires that the particles have extra orbital angular momentum. This softens the divergence, once again rendering the state normalisable.

This same behaviour also occurs in the non-Abelian theory. Roughly speaking, symmetrised representations have anomalous dimensions that scale as $+1/k$, while those of anti-

symmetrised representations scale as $-1/k$. When $k < 0$, it is simple to see that placing too many anyons together in a symmetrised representation will violate the unitarity bound. There is now, however, a similar story for $k > 0$. Perhaps the simplest example of an operator that violates the unitarity bound for $k > 0$ arises in the case of $SU(N)$ with $N_f = N$ different species of scalar, $\phi_a$, each in the fundamental representation. We can then build a baryon operator without the need to add any derivatives: $B = \epsilon_{a_1 \dots a_N} \phi_1^{a_1} \dots \phi_N^{a_N}$. Using the methods above, the dimension of this operator is

$$\Delta_B = N - \frac{N^2 - 1}{2k} \;.$$

This violates the unitarity bound $\Delta_B \geq 1$ for $k < (N+1)/2$. Note that here the bound constrains the rank of the gauge group, $N$. It seems likely that, as in [20], this can once again be traced to the non-normalisability of the quantum mechanical wavefunction.

Interestingly, *effective* non-relativistic theories describing the low energy dynamics of massive relativistic theories always satisfy $|k| > N$ due to quantum shifts of the level. This means that in these theories, the baryon $B$ never violates the bound for any $k$. Nonetheless we can consider non-relativistic theories with arbitrary $k$ for which the bound is non-trivial.

# 4 The fermionic theory

In this section, we give another description of anyons, this time using non-relativistic fermions as the starting point. We will couple these fermions to an $SU(N)_k$ Chern-Simons theory.

The matter consists of $N_f$ complex, Grassmann-valued fields $\psi_a$, each transforming in some representation $R_a$ of $SU(N)$. These fields have non-relativistic kinetic terms, with the action given by

$$S = \int dt \, d^2x \left\{ i\psi_a^\dagger \mathcal{D}_0 \psi_a - \frac{1}{2m} \vec{\mathcal{D}} \psi_a^\dagger \vec{\mathcal{D}} \psi_a - \frac{1}{2m} \psi_a^\dagger F_{12}^\alpha \, t^\alpha[R_a] \psi_a \right\} \;. \tag{33}$$

The coupling to the non-Abelian magnetic field $F_{12}$ plays an analogous role to the quartic interactions in the bosonic Lagrangian (10). (This is particularly apparent if we substitute $F_{12}$ using the Gauss' law constraint.)

Like its bosonic counterpart, this theory also exhibits conformal invariance. The various symmetry generators can be constructed from the number density and momentum current, which are given by

$$\rho = \psi_a^\dagger \psi_a \quad \text{and} \quad \mathbf{j} = -\frac{i}{2}\left( \psi_a^\dagger \vec{\mathcal{D}} \psi_a - (\vec{\mathcal{D}} \psi_a^\dagger) \psi_a \right) \;.$$

The Hamiltonian is given by

$$H = \int d^2x \; \frac{2}{m} \mathcal{D}_z \psi_a^\dagger \mathcal{D}_{\bar{z}} \psi_a \;.$$

As explained in Section 3, we can construct gauge invariant operators by attaching a semi-infinite Wilson line to each particle,

$$\Psi_a(\mathbf{x}) = \mathcal{P} \exp\left( i \int_\infty^{\mathbf{x}} A^\alpha \, t^\alpha[R_a] \right) \psi_a(\mathbf{x}) \;. \tag{34}$$

As before, our interest lies in the spectrum of $n$ anyons in a trap. The most general operator takes the form

$$\mathcal{O} \sim \prod_{i=1}^{n} (\partial^{l_i} \bar{\partial}^{m_i} \Psi_{a_i}^\dagger) \tag{35}$$

where, again, primary operators are those which cannot be written as a total derivative.

However, the anti-commuting nature of $\psi_a$ means that the simplest operators are rather different to those in the bosonic case. Consider, for example, the situation where we have a single species of fermion $\Psi$ transforming in the $\mathbf{N}$ representation of $SU(N)$. Now the operator

$$\mathcal{O} = \Psi^{\dagger n} \tag{36}$$

is non-vanishing only for $n \leq N$ and transforms in the $n^{\text{th}}$ anti-symmetric representation. If we wish to place $n > N$ anyons in a trap, the different operators must be dressed with derivatives. To illustrate this, let's revert to Abelian anyons, charged under a $U(1)$ gauge field. Now there is no operator of the form (36) with $n > 1$. Instead, the operator with the lowest number of derivatives takes the form

$$\mathcal{O} = \Psi^\dagger \, \partial \Psi^\dagger \, \bar{\partial} \Psi^\dagger \, \partial^2 \Psi^\dagger \, \partial \bar{\partial} \Psi^\dagger \dots$$

This operator has $\sim n^{3/2}$ derivatives. At large $k$, this is the ground state of the $n$ anyon system, with $\Delta \sim n^{3/2}$. However, at smaller $k$, the ground state is expected to undergo level crossing.

Computing the spectrum in the fermionic case is no easier than for bosons. Once again, there are two approaches that we can take. The first is the brute force, perturbative approach, valid for large $k$. We describe this below in section 4.1. However, once again there is a class of operators whose spectrum is constrained by their angular momentum. These were described in [20] where they were referred to as *anti-chiral operators* and have only anti-holomorphic derivatives

$$\mathcal{O} = \prod_{i=1}^{n} (\bar{\partial}^{m_i} \Psi_{a_i}^\dagger) \, .$$

For these operators, the dimension is fixed in terms of their angular momentum as

$$\Delta = n + J \, . \tag{37}$$

Note that the opposite minus sign compared to (17). In the context of supersymmetry, these should be thought of as anti-BPS states rather than BPS states.

**Examples**

The simplest example we can consider is a single fermion $\psi$ coupled to an Abelian $U(1)_k$ Chern-Simons theory. This was already discussed in [20]. The simplest anti-chiral $n$-particle operator is

$$\mathcal{O}_n = \Psi^\dagger \, \bar{\partial} \Psi^\dagger \, \bar{\partial}^2 \Psi^\dagger \dots \bar{\partial}^{n-1} \Psi^\dagger \, .$$

This operator has $n(n-1)/2$ derivatives, each of which contributes $+1$ to the total angular momentum. Meanwhile, the angular momentum from the Wilson lines is given by (19) as for the bosonic theory. We have

$$J = \frac{n(n-1)}{2} - \frac{n(n-1)}{2k} \quad \Rightarrow \quad \Delta = \frac{n(n+1)}{2} - \frac{n(n-1)}{2k} \, . \tag{38}$$

In $SU(N)$ gauge theories, the simplest operator (36) sits in the $n^{\text{th}}$ anti-symmetric representation. We have $C_2(\text{Anti-Sym}^n(\mathbf{N})) = n(N-n)(N+1)/N$ and, correspondingly,

$$J = \frac{n(n-1)(N+1)}{2Nk} \, . \tag{39}$$

Recall that for the bosonic case, when $k > 0$ the symmetrised representations increased the dimension of the operator whilst anti-symmetrised ones decreased it. Because of the different sign in (37) relative to (17), this is reversed for fermions.

In the bosonic theories, we saw that certain states violate the unitarity bound. These do not arise in the Abelian fermionic theories, nor in the non-Abelian theories with $k > 0$. However, there are such states in the non-Abelian fermionic theories with $k < 0$, with the baryon being the obvious example.

## 4.1 Perturbation theory with fermions

Non-relativistic conformal fermions with Chern-Simons interactions can be studied perturbatively in much the same way as the scalars in section 3.4. Here we restrict the analysis to one-loop order.

**One-loop corrections**

Similar to the theory with scalars, all one-loop corrections to the operators of the form (35) arise from pairwise diagrams. Therefore to extract the anomalous dimension of such operators we need only to compute the logarithmic correction to the two-anyon operator

$$\partial^{n_1}\bar{\partial}^{m_1}\psi^\dagger_{\rho_1}\partial^{n_2}\bar{\partial}^{m_2}\psi^\dagger_{\rho_2}\,, \tag{40}$$

with the Greek letters $\rho,\sigma = 1,\ldots,\dim R$ denoting the colour indices. (As before, we drop the Wilson lines.) We restrict the analysis to a single flavour of fermions living in the representation $R$ of the gauge group but the generalisation to multiple flavours is straightforward. As in the bosonic case, we focus on the correlation function

$$\langle\psi_{\sigma_2}(p_2)\psi_{\sigma_1}(p_1)\,\partial^{n_1}\bar{\partial}^{m_1}\psi^\dagger_{\rho_1}\partial^{n_2}\bar{\partial}^{m_2}\psi^\dagger_{\rho_2}\rangle\,. \tag{41}$$

At tree level, we schematically denote this correlation function by the following diagram:

$$
\begin{aligned}
&= \delta^{\rho_1}_{\sigma_1}\delta^{\rho_2}_{\sigma_2}(-ip_{1z})^{n_1}(-ip_{1\bar{z}})^{m_1}(-ip_{2z})^{n_2}(-ip_{2\bar{z}})^{m_2} \\
&\quad -\delta^{\rho_1}_{\sigma_2}\delta^{\rho_2}_{\sigma_1}(-ip_{1z})^{n_2}(-ip_{1\bar{z}})^{m_2}(-ip_{2z})^{n_1}(-ip_{2\bar{z}})^{m_1}\,.
\end{aligned} \tag{42}
$$

The only correction this correlation function receives at one-loop arises from the gluon exchange diagram

$$
\begin{aligned}
&= \frac{1}{2k}\log\frac{\Lambda}{\mu}\Bigg[\left(t^\alpha_{\rho_1\sigma_1}t^\alpha_{\rho_2\sigma_2} - t^\alpha_{\rho_1\sigma_2}t^\alpha_{\rho_2\sigma_1}\right)\left(-\frac{i}{2}P^+_z\right)^{n_1+n_2}\left(-\frac{i}{2}P^+_{\bar{z}}\right)^{m_1+m_2} \\
&\quad + t^\alpha_{\rho_1\sigma_1}t^\alpha_{\rho_2\sigma_2}K(p_1,p_2) - t^\alpha_{\rho_1\sigma_2}t^\alpha_{\rho_2\sigma_1}K(p_2,p_1)\Bigg] + \mathcal{O}(\Lambda^2)\,,
\end{aligned} \tag{43}
$$

with $P^\pm = p_1 \pm p_2$. The function $K(p_1,p_2)$ is the same function (27) we encountered in the perturbative study of scalars. The above diagram is sufficient to evaluate the anomalous dimension of operators of the form (35) at one-loop.

**Examples**

Let us start by considering the $U(1)$ theory with a single flavour of fermions. The simplest operator of the form (35) is

$$\mathcal{O}_n = \psi^\dagger\bar{\partial}\psi^\dagger\ldots\bar{\partial}^{n-1}\psi^\dagger\,. \tag{44}$$

In the supersymmetric theory this is an anti-chiral primary operator and is therefore one-loop exact. This holds true even in the non-supersymmetric theory and the operator is only corrected by the pairwise diagrams correcting $\bar{\partial}^{m_1}\psi^\dagger\bar{\partial}^{m_2}\psi^\dagger$ which evaluate to

$$\text{(diagram)} = \frac{1}{2k}\log\frac{\Lambda}{\mu} \quad \text{(diagram)} \quad + \mathcal{O}(\Lambda^2).$$

As this is independent of the number of derivatives $m_i$ the dimension of $\mathcal{O}_n$ is simply

$$\Delta = \frac{n(n+1)}{2} - \frac{n(n-1)}{2k}\,, \tag{45}$$

as derived earlier (38). We remark that – as was observed in [20] – the spectrum of $\mathcal{O}_n$ in the $U(1)_k$ theory with fermions matches precisely that of $\tilde{\mathcal{O}}_n = \phi^{\dagger n}$ in the $U(1)_{\tilde{k}}$ theory with scalars if we choose $1/\tilde{k} = 1 - 1/k$.

Another important example is the baryon operator in $SU(N)$ Chern-Simons theory

$$B = \psi_1^\dagger \ldots \psi_N^\dagger\,. \tag{46}$$

More generally, we can consider the operators

$$\mathcal{O}_{\rho_1\ldots\rho_n} = \psi_{\rho_1}^\dagger \ldots \psi_{\rho_n}^\dagger\,, \tag{47}$$

with $B = \mathcal{O}_{1\ldots N}$. The pairwise diagrams that contribute to the anomalous dimension of these operator evaluate to

$$\text{(diagram)} = -\frac{N+1}{2Nk}\log\frac{\Lambda}{\mu} \quad \text{(diagram)} \quad + \mathcal{O}(\Lambda^2)$$

The dimension of the $\mathcal{O}_{\rho_1\ldots\rho_n}$ therefore evaluates to

$$\Delta = N + \frac{n(n-1)(N+1)}{2Nk}\,. \tag{48}$$

consistent with (39).

## 4.2 A comment on bosonization

In previous sections we have studied Chern-Simons theories coupled to both bosons and fermions. Yet, in both cases, the resulting particles are neither: they are anyons. This motivates the possibility that the theory of bosonic and fermionic theories are actually equivalent.

Indeed, there is now a well established set of conjectures relating relativistic, bosonic and fermionic Chern-Simons matter theories. These typically go by the name of non-Abelian bosonization dualities and, roughly speaking they relate $SU(N)_k$ bosonic theories to $U(k)_N$ fermionic theories [28–31]. More precise statements of the dualities at finite $N$ and $k$ were given in [32–34], and various pieces of evidence for the duality were complied in [35–42].

It is interesting to ask whether there is a non-relativistic counterpart of these dualities. Evidence was given for the equivalence of quantum Hall states in [34] when $N_f = N$ and one can ask if this extends to the spectrum. Naively, however, this does not appear to be the case. The kind of operators (13) and (34) that we have discussed above are gauge invariant by virtue of the attached Wilson line which extends to infinity. But they nonetheless transform non-trivially under the global part of the gauge group. For such states, the two theories most certainly are not equivalent. Perhaps the simplest way to see this is that they have different global symmetry groups, $SU(N)$ and $U(k)$ respectively. Nonetheless, there is clearly a similarity between the spectra of the bosonic and fermionic theory described above. It would be interesting if this could be extended to a full bosonization duality in this non-relativistic context.

(Note added: As this paper was being prepared for publication, this issue was addressed in [43].)

## Acknowledgements

Thanks to Guy Gur-Ari for useful discussions. DT and CT thank TIFR and Stanford Institute for Theoretical Physics for their kind hospitality while this work was undertaken. We are supported by STFC and by the European Research Council under the European Union's Seventh Framework Programme (FP7/2007-2013), ERC grant agreement STG 279943, "Strongly Coupled Systems".

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
