# Peer review of "The Conformal Spectrum of Non-Abelian Anyons"

_SciPost Physics, doi:SciPost Phys. 4, 022 (2018)_

## Round 1 · Referee Report · Anonymous (Referee 1) · 2017-4-28

Strengths

1. The authors make a clear connection with the abelian three-anyon problem
2. The introduction is well written
3. Quantum mechanical conformal invariance is well-explained
4. The authors use this invariance to determine the spectrum of non-abelian anyons
5. As an alternative, they use perturbation theory to determine the spectrum of some of the states

Weaknesses

1. The authors should include a plot like figure 1 to illustrate the difference between abelian and non abelian anyons.
It would be nice to see which states can be calculated analytically and which ones are not.

2. It is not clear which fraction of the states are accessible analytically. Could the authors comment on that?

Report

This is a well written paper that should be published after including the suggestions below.

Requested changes

1. Add a figure as Fig. 1 for the case of three nonabelian anyons (with only the analytical states).
2. Explain how the nonabelian action affects the number of states that can be obtained analytically.

---

## Round 2 · Author Response

We thank the referee for their comments.
The referee raises an interesting question: what fraction of the states in the system are under analytic control? Unfortunately this appears to be a rather difficult problem and we do not have the general solution. Indeed, even for seemingly simple cases, like 3 anyons in SU(2), it is complicated.
In an attempt to improve the paper along the lines suggested by the referee, we have included a plot of the low lying states of two SU(2) anyons, in which we include both analytic and non-analytic states. We have elaborated on the difficulty in counting the analytic states in general.
The referee raises an interesting question: what fraction of the states in the system are under analytic control? Unfortunately this appears to be a rather difficult problem and we do not have the general solution. Indeed, even for seemingly simple cases, like 3 anyons in SU(2), it is complicated.
In an attempt to improve the paper along the lines suggested by the referee, we have included a plot of the low lying states of two SU(2) anyons, in which we include both analytic and non-analytic states. We have elaborated on the difficulty in counting the analytic states in general.

---

## Round 2 · List of Changes

The changes are all on page 21. The referee requested a plot of analytic states for 3 anyons to match that in the introduction. While this plot is simple to construct, we felt that it didn't add much without knowing the non-analytic states as well. This is why we plumped for two anyons instead.

---

## Editorial Decision

published